# Polish Transition towards Circular Economy: Materials Management and Implications for the Construction Sector

**DOI:** 10.3390/ma13225228

**Published:** 2020-11-19

**Authors:** Justyna Tomaszewska

**Affiliations:** Instytut Techniki Budowlanej, Ksawerów 21, 02-656 Warsaw, Poland; j.tomaszewska@itb.pl; Tel.: +48-5664-343

**Keywords:** circular economy, Poland, life cycle assessment, recycling and reusability, ecofriendly solutions, natural resources consumption, construction

## Abstract

Poland’s economy as the sixth largest economy in the EU is painfully experiencing the effects of natural resource depletion, which extend to increasing prices and the growing dependence on foreign suppliers. The situation is particularly unfavorable in the construction sector, which is among the most resource- and energy-consuming areas of the economy. This paper juxtaposes the situation of Polish construction industry in the context of the national economy with the context of the evolving EU policies promoting green solutions. The resulting changes in Polish legislation, industry and society are identified. The implementation of selected Circular Economy (CE) aspects, outcomes, emerging challenges and future directions are discussed. The social aspects related to this transformation were analyzed based on a survey carried out among construction industry consumers. The results clearly highlight that individuals are aware of the need to protect the environment, but also indicate a strong necessity to educate the populace about the consequences of the excessive exploitation of the environment and the long-term benefits of CE solutions.

## 1. Introduction

The implementation of essential changes in the way a domestic economy operates is a complex process. The transformation of the linear economy towards the Circular Economy (CE) is a challenge that many European countries, including Poland, currently face. Changes in product management at the final stage of their life cycle are what differentiate the linear economy from the CE. The essence of the CE concept for raw materials is to prevent waste generation, e.g., by extending a product’s life cycle or creating possibilities for repair [1,2]. When it is not possible to avoid waste generation, the product should be suitable for reuse or recycling, according to the waste management hierarchy shown in Figure 1 [3]. Intensive industrial development and the ever growing global population, which at present exceeds 7.8 billion and is forecast to reach 9 billion by 2037 and 10 billion in 2050 [4] are among the key factors responsible for the increase in the rate of natural resources consumption. This is encapsulated in the Earth overshoot day (EOD), which is the day when all natural resources, which could renew in a given year, are exhausted. In 2019, the EOD was on 29 July, which was the soonest in the history of humankind. In the EU, the situation was worse, with overshoot day occurring on 10 May and in Poland on 15 May. By comparison, Qatar’s EOD was on 11 February, while in Kirgizstan it was 26 December [5]. The volume of natural resource consumption in the EU, in 2014, was estimated at 7.4 billion tonnes (3.1 billion tonnes were used for power generation), of which 5.8 billion tonnes were from local mining, 0.9 billion tonnes were net import and only 0.7 billion tonnes were secondary raw materials [6]. OECD expects the global consumption of natural resources to double by 2060 compared to 2011 (when it amounted to 79 billion tonnes), if the current consumption rate is maintained. The emission of greenhouse gases is predicted to increase in the same period from 28 to 50 billion tonnes of CO_2_ equivalent [7].

The Polish economy as the sixth largest EU economy, with its GDP per capita amounting to 71% of the EU average in 2018 [8], painfully experiences the effects of natural resource exhaustion, which includes rising prices and increasing dependence on foreign suppliers. The situation is particularly onerous in construction, which is among the most resource- and energy-consuming economic sectors, and which has already been struggling with severe labor shortages for the past few years. The domestic consumption of raw materials reveals an increasing trend, whose dynamics vary depending on the material group. For instance, between 2015 and 2018 the domestic consumption of non-alloy aluminum went up by 97.7%, plastics by 29.9%, float glass by 28.2%, metallurgical products by ca. 38.0%, wood-based materials (chipboard) by 25.6%, cement by 21.7%, and paper and cardboard by 10.0% [9], while the construction industry reported an increase as high as 23.2% in construction and assembly outputs [10]. 

With regard to the above, closing the circulation of raw materials in the economy seems to be the only way towards hampering natural environment devastation as a result of anthropogenic actions, especially in construction [11,12]. The concept of the CE assumes the existence of an industrial economy which is renewable by assumption; its supreme objective is to ensure the effective flow of raw materials, power, labor and information so that it is possible to “rebuild” environmental and human resources [1,13,14,15]. The EU predicts that the implementation of CE concepts will help to reduce CO_2_ emissions by 450 million tonnes by 2030, making 600 billion EUR in savings for enterprises (8% of annual turnover) and establishing 580,000 new jobs [16]. Therefore, it seems obvious that reaching the aforementioned targets requires a holistic approach, which engages actors across the supply chain at all stages of a product’s life cycle. This paper presents the characteristics of the Polish construction industry vis-à-vis the domestic economy and the continuously evolving EU policies promoting green solutions, and the consequent changes in Polish legislation, industry and society. The implementation of selected CE aspects, outcomes, emerging challenges and future directions are discussed. Social aspects related to the changes being implemented were analyzed based on a survey carried out among construction industry consumers. The results confirm that the respondents were aware of the need to protect the environment, but also emphasize the strong need to educate society about the consequences of overexploitation of the environment and the long-term benefits of CE solutions. Despite a plethora of studies on CE implementation in Europe and the individual economies of member states, none so far refers specifically to the Polish situation in any detail. This paper fills this gap by presenting the progress of EU countries’ economic transformation, especially in dynamically developing post-communist countries.

## 2. Materials and Methods

In order to fully understand drivers and the adopted mechanisms of Polish transformation to the CE model, as well as the current situation of the construction sector, comprehensive research has been performed—including literature reviews (special emphasis on Polish and EU regulations and statistics data), symposia, interviews and surveys. Conclusions presented during the series of debates and symposia organized by the Institute of Circular Economy (IGOZ), Ministry of Development, National Fund for Environmental Protection and Water Management (NFOŚiGW), The Institute for Innovation and Responsible Development (Innowo), Association of Polish Architects (SAPR) and the Polish Association of Engineers and Construction Technicians (PZITB) allowed a summary of the current status of the process, to define research questions and hence to build a database of individuals to be interviewed. The main obstacles of the transformation process have been defined based on the personal discussions with the representatives of academics, private construction, policy makers and government officials. The general degree of construction consumer awareness in regard to environmental protection, and their perception of circular solutions has been determined in the process of a survey carried out among young- and middle-aged Poles (305 respondents) by means of e-mails and social media (Facebook) in the period of November 2019–January 2020 (Table 1).

## 3. Polish Legislation versus EU Measures

The Seven Basic Requirements for Construction Works of Construction Product Regulation (CPR no 305/2011) was a milestone towards the sustainable use of natural resources in EU construction. The document states that construction works must be designed, built and demolished in such a way that the use of natural resources is sustainable and in particular ensure the following: (a) reuse or recyclability of construction works, materials and parts after demolition; (b) durability of construction works and (c) use of environmentally compatible raw and secondary materials in construction works [17]. In December 2015, the European Commission adopted the Circular Economy Package [18], which included a plan for CE implementation in member states [3] (updated at the beginning of 2019 [19]). The plan outlines a number of actions related to the establishment of policy and mechanisms, which enable the maximum service life of raw materials already circulating in the economy, reduction in the amount of generated waste, and the manufacturing of new products, which are more environmentally friendly and designed with consideration of their life cycle. In 2011, the Single Market for Green Products Initiative (SMGP) [20] was established, aimed at developing uniform conditions of the environmental efficiency measurement of products and services available on the EU market. Consequently, in 2013, recommendations were published (2013/179/UE) [21] on the use of common methods of product environmental footprint (PEF) [22] and organization environmental footprint (OEF) [23] measurements. PEF and OEF are based on existing methods and standards, and were created as part of a cross-sectoral approach. The pilot phase spanned over three years (2013–2016), whereas the current phase constitutes a transition before the potential adoption of the policy implementing PEF and OEF [24]. The implementation of CE requires a systematic analysis of progress and emerging challenges, and thus the “Key indicators for a monitoring framework” was developed, covering ten indicators [25], which describe the changes in the economy and which are based on official statistical data (e.g., from Eurostat, Joint Research Centre and the European Patent Office).

Poland, as a member state and in response to the actions taken by the European Commission towards CE implementation, in February 2017, adopted the Strategy for the Country’s Responsible Development by 2020 with a perspective by 2030 [26] (updated Strategy of the Country’s Development 2020 of 2012 [27]), which contains the general vision of the country’s development and a strategic project called Roadmap of transformation towards CE [28]. State administration bodies in agreement with public and private sector institutions, as well as other member states, developed best practices for the implementation of CE and new domestic standards, as part of such projects as R2π [29] and “otoGOZ” [30]. It should be noted that CE elements have been implemented for at least a few years now as part of programs operating as: the green economy [31], cleaner production [32], sustainable development [33] and low-emission economy [34], all of which contribute to raw material circulation capping. Their aim is to make and use products in the most efficient way, and to manage the resulting waste in an economically and environmentally optimum way.

## 4. Circular Economy Roadmap

The draft Polish CE roadmap, adopted in September 2019 [28], is based on a model which assumes the coexistence of biological nutrients, designed to re-enter the biosphere safely and build natural capital, and technical nutrients, which are designed to circulate at high quality without entering the biosphere [1,35]. A dramatic decrease in the demand for primary raw materials, with a simultaneous reduction in the number of pollutants and generated waste are expected as the consequences of closing the biological and engineering loop. The actions included in the CE Roadmap are mandatory on a national level and foster the execution of four Polish priorities related to CE implementation [3], namely: (a) innovativeness, strengthening cooperation between the industry and science, and consequently the implementation of innovative solutions in the economy; (b) creating a secondary raw materials European market; (c) providing high-quality secondary raw materials; (d) service sector development. The provisions concerning the following measures are particularly important for the construction industry:Analysis of potential and legislative changes proposed to increase the economic use of incineration by-products. Over 78% of electrical energy in Poland is generated from lignite and hard coal. During incineration of these raw materials, however, gases and incineration by-products are generated, including ash, slag and dust. Incineration by-products are the source of many minerals whose potential is not fully exploited. Owing to their binding characteristics, they can be used for the production of construction materials, mainly cement and concrete (as Portland cement substitute), and in road and underground construction as an ingredient of embankments and other structural layers [36,37].Creating a platform dedicated to secondary raw materials, aimed at market actors, providing information about supply and demand, and trade.Extended producer responsibility (EPR), encouraging producers to design and make products with extended service life, and imposing the obligation to collect and manage waste from the same products as the ones launched.Identification of the environmental impact of products and services using, for example, the Life Cycle Assessment (LCA), Product Environmental Footprint (PEF) and Organization Environmental Footprint (OEF) methods [38].Identification of all streams of municipal waste, including post-consumer waste, previously not recorded but having an economic impact. This measure can be particularly useful in the context of the classification of waste (effective until 2018) generated during minor repairs, construction and demolition works by individuals, incorporating construction and demolition waste into the stream of municipal waste.Changes in public procurement law, which would generate demand for products and services according to CE business models and increasing the share of green public procurement.Analysis of the possibilities to implement reporting and inspection reliefs guidelines for entities that observe environmental standards, such as the EU Eco label and EMAS, and entities entered in the Polish Register of Cleaner Production and Responsible Entrepreneurship.The systematic assessment of the Polish economy transformation towards CE and the impact of the new economy model on the social and economic development of Poland will be carried out with two sets of indicators developed by state administration bodies in cooperation with public sector institutions and industry representatives as part of the “otoGOZ” programme [30].

## 5. The Construction Sector in the Context of the Domestic Materials Economy

The construction sector is an essential component of the Polish economy. Its share in the GDP is estimated at 7.0% [39]. A characteristic feature of this sector is its highly diversified development dynamics. This is mainly due to the coexistence of a fairly small group of medium and large quickly developing businesses, responsible for the execution of large construction investments, and a large group of small, often highly specialized companies doing repair works or small construction projects commissioned by natural persons or subcontracted by medium or large enterprises (no of employees >9). The value of construction and assembly production in Poland in 2018 amounted to 52 billion EUR, 94.4% of which originated in sales of construction works by construction companies using their own resources (51.7% was generated by microenterprises with a workforce of <9 people). The remaining 5.1% was production executed by non-construction companies with a system of orders and construction works structured according to the “do-it-yourself” system, i.e., by enterprises and natural persons for their own needs. The construction and assembly output in 2018 saw a 14.3% increase year-to-year and an increase in the range of 23.2% and 31.8% compared to 2015 and 2010 respectively. Over 97% of the construction enterprise production belongs to private sector units, while the share of the public sector did not exceed 2.1% between 2005 and 2018. The total value of the Polish construction works exports (enterprises with >9 employees) is estimated at 1.5 billion EUR, nearly half of which is executed in Germany (Table 2) [8,10,40].

Poland is a producer of a large number of materials and products, which fully satisfy the demand of the domestic market and which are exported. The group of exported products includes paper and paperboard, plastics, steel products (steel pipes, hot-rolled products), sulfur, some non-ferrous metals, fiberboard and synthetic rubber. At the same time, Poland imports many raw materials (e.g., cotton, natural rubber, tin) due to lack of domestic resources, insufficient domestic production capacity (e.g., aluminum, wool, chemical fibers, some plastics and steel products), as well as consumer preferences. Poland’s structure of material consumption indicates that the industry is the largest consumer, responsible for 70–100% of total consumption. The construction business is the most important consumer of cement, lime, roofing paper, steel bars and profiles, steel pipes, galvanized steel sheets, cables and wires, and some aluminum and plastic products. A dynamic increase in the construction and assembly outputs is among the driving engines of the domestic industry, which also generates greater demand for natural raw materials and prefabricated products. Between 2015 and 2018, the consumption of non-alloy aluminum increased by 97.7%, plastics by 29.9%, float glass by 28.2%, some steel products by ca. 38.0%, wood-based materials by 25.6%, cement by 21.7%, and paper and paperboard by 10.0% (Figure 2) [9,40]. The effects of the overexploitation of the environment and the “take-make-dispose” linear economic model followed by most countries, including Poland, are related to the depletion of natural resources, which contributes to the increase in prices and the ever greater dependence on foreign suppliers. When analyzing the share of each component in the end cost of selected materials and products commonly used in construction (Table 3), one can observe that over 50% of costs in Poland are related to the purchase of raw materials and energy. Combined with workforce shortages caused e.g., by the migration of Polish specialists to western Europe because of more favorable financial conditions, the maintenance of the current dynamics of Polish construction development is jeopardized. This becomes even more serious taking into account Poland’s power mix, where hard coal and lignite are the basic source of power (74%), whereas the share of renewable energy sources (RES) amounts only ca. 13%, despite its gradual increase over the past few years [41].

A positive trend observed in the national materials economy over recent years is the greater importance of secondary raw materials and recycled materials. Many materials of mineral and organic origin, such as natural rubber, wood and paper return to production as secondary raw materials. Greater significance is also attributed to the reuse of waste from production and purchase due to environmental and economic reasons [43,44,45]. The period between 2015 and 2018 saw a significant increase in the acquisition of selected groups of waste by production and trading entities. Steel and cast iron scrap, as well as waste paper and paperboard, make up the largest groups of acquired waste (Figure 3).

The pronounced diversity of construction makes it difficult to develop one valid path of transition to the CE model. The Institute for Innovation and Responsible Development (Innowo), which is a non-government organization supporting the development of innovation and implementation of system changes for sustainable social and economic development analyzed various obstacles that prevent the implementation of CE concepts in the Polish construction industry. Innowo emphasized the existence of unfavorable investment financing models, the lack of the economies of scale effect, regulations that are adopted too hastily, lack of waste stream tracing, the wider perceptions of reuse of building materials and the limited possibilities for recycling them [46].

## 6. Construction and Demolition Waste in Domestic Waste Management

In Poland, waste is classified based on the catalogue of waste included in the regulation of the Ministry of Climate [47]. The Catalogue covers ca. 950 kinds of waste classified in 20 groups. A total of 128 million tonnes of waste were generated in 2018, 9.8% of which was municipal waste. The quantity of waste generated between 2000 and 2018 (excluding municipal waste) remained stable at 110–130 million tonnes a year (115 million in 2018). Analyzing the dynamics of the change in the quantity of generated waste against the GDP—5.1% increase in 2018 versus 4.8% in the previous year—a positive trend can be observed, because the rise in GDP matches the stabilization of the generated waste level [48]. According to data from Eurostat, the municipal waste index in Poland per capita is among the lowest in the EU: in 2018 it was 329 kg (28% of which was collected selectively), whereas the EU-28 mean was 488 kg. Mining and quarrying are the main sources of waste in Poland, followed by industrial processing and power generation and supply (Table 4).

Waste from construction sites, repairs and demolition of building structures and road infrastructure (including soil from contaminated areas) are classified as group 17 in the Catalogue of Waste. By 2018, any waste generated during minor repairs, construction and demolition works, not governed by the regulations of the Building Law Act [49] and performed by individuals, was classified as construction and demolition waste (CDW) in the stream of municipal waste (group 20 municipal waste, including selectively collected fractions), while hazardous CDW was classified as group 17. The EU Directive 2018/851 [50] adopted in 2018, amending the framework directive on waste (EU) 2008/98 [51], defined CDW in greater detail, including all CDW from minor and individually performed construction and demolition works in households in group 17 in the Catalogue of Waste. Currently, there are two common disposal paths of CDW generated by individuals: one includes the submission of waste free of charge to peripheral Municipal Selective Waste Collection Points (PSZOK)—Figure 4b, which are often located within the premises of the Municipal Services Department. This solution is very popular in towns and villages, where the PSZOKs are located fairly close to the waste generation source, encouraging people to submit personally their waste to the Collection Point. The solution most popular in towns and cities is to order a container for CDW (usually between 1 and 20 m^3^) from a private company—Figure 4a, which transports waste to a landfill or from a company which manages and recovers waste. The current system of CDW submission and collection has not fully eliminated the problem of illegal landfills, but has definitely helped to reduce the scale of the phenomenon to single-digit incidents in communes per year.

Concrete and brick debris form one of the largest groups of CDW. This waste is mainly processed by crushing and sorting, which enables its reuse, for example as aggregate in the construction of roads, embankments, railway embankments, the production of concrete mixes or land hardening [52,53]. The net price per tonne of crushed debris in Poland ranges from 2.3 to 6.8 EUR, while the cost of crushing ranges from 2.9 to 4.5 EUR per tonne [54]. In the case of plastics, namely expanded polystyrene (EPS), commonly used for heat insulation in Poland, a technology developed as part of the European PolyStyrene Loop (PSLoop) project [55] is expected to offer considerable opportunities for managing the large volumes of this type of waste. This technology enables the separation of harmful additives. The operation of two plants based on this technology is planned in Poland [56]. 

CDW is the largest stream of waste in the EU, both in quantity and volume, constituting 36% (ca. 924 million tonnes) of all generated waste. On average, 46% of CDW is recovered [52]. An average inhabitant in Europe is estimated to generate no less than 160 tonnes of CDW in their lifetime. As a result of EU environmental policy, the mandatory levels of recovery and recycling of each stream of waste have been gradually raised [50,57,58,59]. The Polish waste recovery rate, for waste generated during the construction and demolition of buildings, is estimated at 91% [8], which fulfils the requirement of 70% recovery in all EU countries by the end of 2020 [51].

## 7. Obstacles on the Road of Polish Construction to CE

The process of implementation of new construction solutions is fairly long and demanding, which is primarily caused by constraints resulting from user safety issues. Accordingly, construction takes advantage of proven solutions that have been marketed for at least a few dozen years, contributing to the minimal increase in the sector’s labor productivity, which has been stable at 1% a year over the last 20 years [40,60]. Taking also into account the pronounced diversity of the sector, caused by the coexistence of a small group of quickly developing medium and large enterprises, often with high equity, and a numerous group of microenterprises with narrow specializations and limited investment, the development of a universal path of transition to the CE model poses serious challenges. One should note, however, that some features such as durability and invariability, or functionalities including the capacity to perform maintenance and repairs, and the adaptation, e.g., of buildings are synonymous to products made by the construction sector. At the same time, they form the foundations of CE. Based on that, it can be concluded that in the transformation of the economy to the circular model, construction stands in a privileged position. Nonetheless, with regard to the comprehensive nature of this process, which engages all industry branches and actors across the whole of the value chain at all stages of a product’s service life, there are many barriers emerging on the road of construction to CE. This section of the paper presents the factors which hamper the implementation of CE in Polish construction, with a special consideration of disturbance of relational and logistic systems market barriers technological context and social aspects.

### 7.1. Disturbance of Relational and Logistic Systems

The lack of a holistic approach, integrating actors and actions at all stages of a building’s life cycle is among the basic organizational barriers related to the planning and implementation of CE solutions in Polish construction. With regard to the increasingly stricter EU limits pertaining to waste storage and heat processing, the greatest efforts focus on waste management, including recovery, reuse and recycling. The issues related to the development of structural solutions, which ensure the appropriate waste re-streaming, however, are neglected. Consequently, despite the fact that on the Polish market there are businesses offering waste management practices compliant with CE, their capabilities are limited due to the insufficient quantity of resources (waste) submitted. The consequences of this can be analyzed from at least two points of view. First of all, this prevents the maximum service life of the materials found in a product and, hence, their long-lasting presence in the economy, which contradicts the basic assumptions of CE. Furthermore, the occurrence of such a phenomenon discourages other units from taking action to improve existing solutions and develop new ones. 

Another barrier emerging in Polish reality is the lack of common use of “green” requirements in public tenders. Back in 2008, the European Commission recommended that member states apply uniform criteria in public tenders aimed at improving environmental conditions [61]. In light of the Polish Public Procurement Law regulations, effective as of 2016, the contracting entity selects the most favorable bid according to the criteria assumed, referring to the best price or building cost criterion, calculated using the life cycle costing (LCC) methodology [62]. According to a report from the Public Procurement Office, in 2017, there were 139,133 procurements awarded of a total value of 163.2 billion PLN (ca. 8.23% of GDP), with less than 1% (ca. 2% of the value) of the procurements addressing environmental and innovation aspects. The LCC of a building was a bid assessment criterion applied in only 17 procurements [63].

### 7.2. Market Barriers

The lack of structural solutions for directing waste stream fractions to the relevant recipients—users or recyclers—frequently causes uncertainty in terms of the material resources (waste) supply continuity. At the same time, with regard to fairly small production sizes, reaching the economies of scale effect becomes practically impossible, and in many cases leads to an absurd situation in which the secondary raw material price is higher than the price of the same raw material acquired directly from the environment. Consequently, building materials and process solutions manufactured according to CE rules become non-competitive compared to conventional materials. The situation is additionally aggravated by the aforementioned price prioritization in the decision-making process of investors who are not the future users of construction objects. The lack of public subsidies for the production of secondary raw materials and products manufactured from them could probably be compensated by the mandatory application of LCC for the whole life cycle of a building (green public procurement) and the introduction of positive financial stimuli, such as tax exemptions for buildings holding green certificates confirming their environmentally friendly character. 

Poland also lacks the market mechanisms and business models, which promote the “life extension” of products no longer needed by their owner or which encourage the use of secondary raw materials and products that can be recycled at their end of life. There are also no financial instruments providing temporary protection for newly established businesses willing to implement circular concepts in the Polish economy.

### 7.3. Technological Context

Separation between the role of investor and future user in the construction process has become common practice in Poland, especially in the public sector. This phenomenon is perceived as a major obstacle for the development of sustainable civil engineering. The simultaneous strive of both parties to pursue their specific and often contradicting goals often leads to solutions which do not comply with the fundamental values of CE. With regard to the interests they represent, investors usually focus only on the building construction stage and attempt to maximize their profits by minimizing expenditure. If there are no clear guidelines, e.g., relevant laws and regulations, product price and not functionality plays the key role in the decision-making process. This may affect the building’s environmental impact and its life cycle, which are in turn extremely important from the future user’s perspective. This situation is additionally exacerbated by the dynamic increase in the construction and assembly outputs observed in Poland over the last few years, and the rising imbalance between demand and supply, caused mainly by an increase in the price of raw materials, construction products and labor shortages. Consequently, investors try to reduce construction time, which is often inversely related to the total durability of the structure. 

Another obstacle to CE in construction is the direct result of essential characteristics of construction materials and products created many years before, namely their durability which determines service life. Often, such materials contain chemicals that are currently banned, which precludes the possibility of processing their waste. For example, hexabromocyclododecane (HBCD), commonly used as a flame retardant for expanded polystyrene (EPS) products was forbidden in the EU after 21 August 2015. Other examples include orthophthalates (DEHP, DBP, BBP and DIBP) used as plasticizers in polyvinyl chloride (PCV) products, such as floor tiles and carpets, shower curtains or siding tiles, which were entered on the REACH list in December 2018. Consequently, it is impossible to use such products, and the development of dedicated recycling technologies is unprofitable. Nonetheless, there are exceptions, such as the aforementioned EPS waste processing technology developed as part of the European PolyStyrene Loop (PSLoop) project [55], which allows the separation of harmful additives. Two plants operating on this technology are planned in Poland [56].

### 7.4. Social Aspects

The transformation to CE implies the need to develop product and service innovations, creating new organizational mechanisms, business models and financial instruments whose effectiveness, regardless of the degree of their refinement, will measure social approval expressed as increased demand for products made according to CE principles. It is evident that no changes can be effectively implemented without consumer involvement, especially when it comes to far-reaching changes such as the transition to another economic model. For the purposes of this paper, the author carried out a survey in order to discover the degree of construction consumer awareness in regard to environmental protection, and their perception of circular solutions. The study was carried out by means of e-mails and social media (Facebook). The study population included 305 Poles (147 females and 158 males), representing three age groups: 20–39 years (84.6%), 40–59 years (12.1%) and 60+ years (3.3%); 89.9% had completed their higher education; 9.2% had completed secondary education and less than 1% primary education. Representatives of the construction industry constituted 8.9%, while 52.8% of the respondents declared no relationship with the building sector. The results of the survey are presented in Table 5. Society’s fairly low awareness of the consequences of overexploitation of the natural environment was identified as a major barrier for the implementation of CE solutions in many countries [46,64,65,66,67]. Similar conclusions can be drawn from a survey in which over two thirds of the respondents declared sufficient knowledge of environmental protection issues, while only 14% revealed a similar level of knowledge of sustainable construction.

The consequences of fairly limited consumer knowledge of sustainable development and CE issues can be found in the hierarchy of criteria applied when purchasing building materials what is presented in Figure 5. Every second respondent in the study mentioned price as the main selection criterion (multiple choice question), while only 28% of the survey participants took environmental aspects into account.

The results of studies conducted in the UK revealed that consumers’ negative approach to building materials and products manufactured from recycled raw materials is the major factor preventing their common use [64]. Hence, a conclusion can be drawn that economic transformation can be successfully and efficiently implemented if the long-term benefits and consequences of using CE-compliant solutions are communicated to consumers. This paper presents only some of the results of the survey. Other results can be communicated by the author upon request.

## 8. Conclusions

Circular economy solutions help to maintain the added value of products and eliminate waste. The possibility of the reuse of natural resources after the end of a product’s life cycle supports their effective maintenance in economic circulation and, hence, limits their negative environmental impact [43,44,45].The effective implementation of CE assumptions requires changes across the whole of the value chain, from the stage of design and new business model development to innovative methods of waste collection and processing, and the development of new consumer habits. It also entails the need to reorganize the system and introduce numerous economic, organizational, social and political innovations.Regardless of the acquired strategy, some elements of linearity will not be fully eliminated, mainly because of the need to use resources that have not been used previously or the generation of unavoidable residual waste.The implementation of CE solutions is estimated to contribute to the reduction in EU demand of primary resources by 17–24% by 2030, which will bring in savings in the European industry of around 600 billion EUR a year [2,16].Poland’s transition to the CE model implies the need to develop product and service innovations, the development of new business models, organizational charts and, most importantly, to increase social awareness. In order to maximize the benefits of transformation and theoretical elimination of the risks related to CE, the changes being implemented should also be coordinated on a legislative, organizational and business level. In construction, it is important to develop instruments that allow an overlap between investor and user goals.The promotion of circular solutions by authorities should go along with considerations for technological constraints and the provision of maximum support for the development of product and service innovations, including consumer safety.It is also important to remember that regardless of the refinement of the implemented solutions and their expected engineering effectiveness, the end user will ultimately remain the final judge of the efficiency of any changes. Society’s response can either impede or trigger the implementation of CE solutions. The conclusions drawn from the survey reveal that Polish society is aware of the need to protect the natural environment, but that certain CE aspects have not yet been fully discovered, despite information frequently shared in the media. Therefore, it can be expected that increased efforts in educating society on the consequences of environmental overexploitation and the long-term benefits of CE solutions can be key in the effective economic transformation of Poland.In addition to the obvious environmental and economic benefits resulting from the implementation of the CE model, one should remember that social benefits, including the trend of creating new jobs, is forecast in many analyses [16,68,69]. Nevertheless, the real impact of actions taken should be should be systematically monitored. In consequence, detailed analysis of associated changes occurring in the particular branches of the economy with the special emphasis on the construction sector and the whole supply chain can be proposed as evident continuation of this research. The global dimension of the transformation should be also considered since it directly impacts on the international politics of Poland.

## Figures and Tables

**Figure 1 materials-13-05228-f001:**
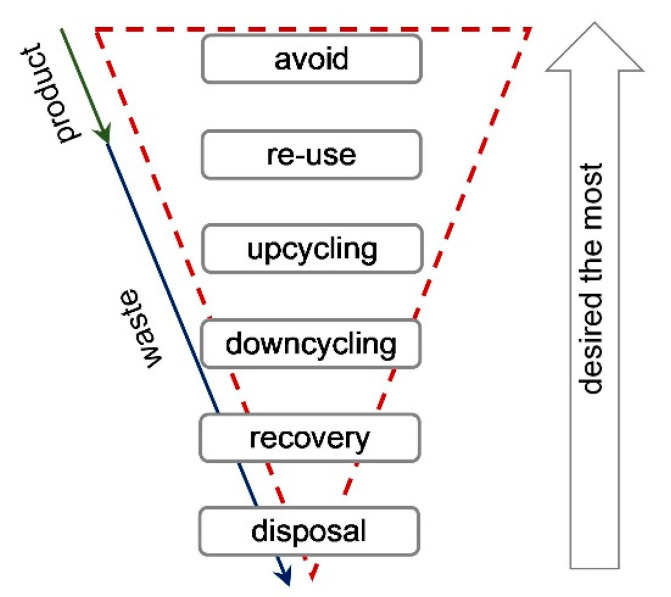
Waste management hierarchy in the circular economy (CE) [3].

**Figure 2 materials-13-05228-f002:**
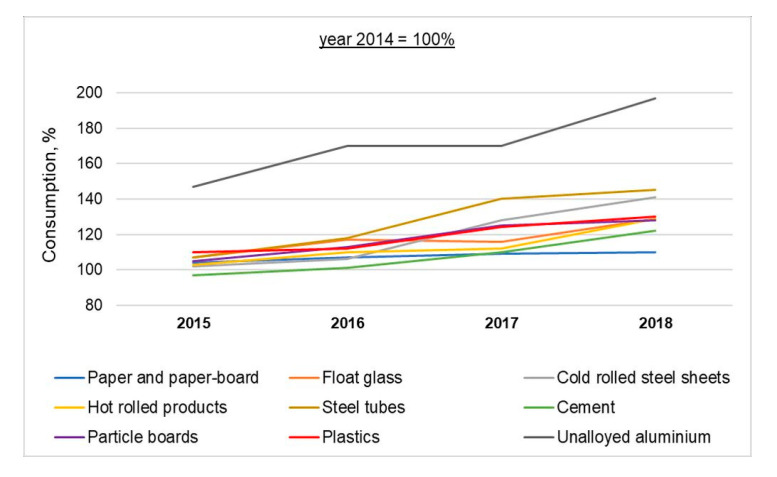
Domestic consumption of selected materials between 2015 and 2018 [9].

**Figure 3 materials-13-05228-f003:**
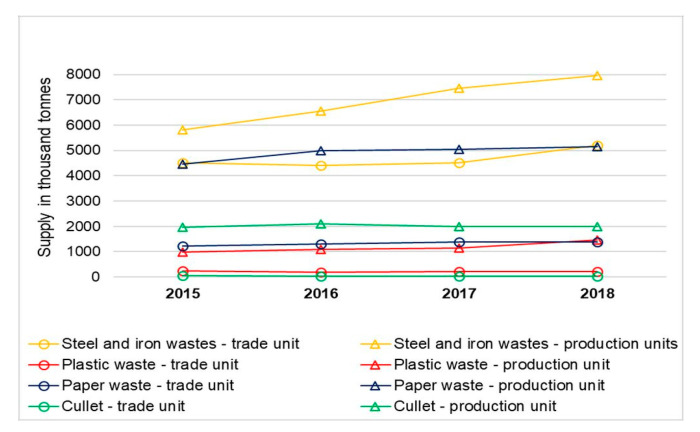
Supply of steel and iron waste and selected non-metallic waste suitable for recycling in production and trade units in 2015–2018 [9].

**Figure 4 materials-13-05228-f004:**
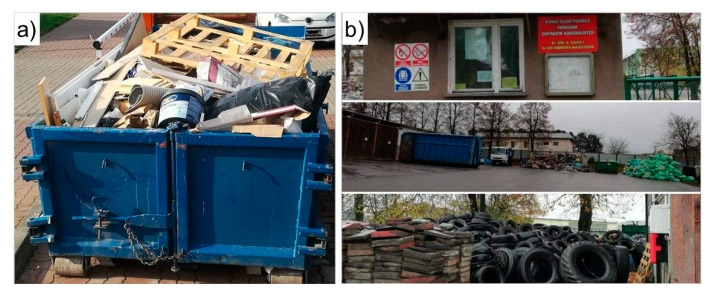
Examples of construction and demolition waste (CDW) collection methods in Poland: (**a**) a container placed near a renovation site in Warsaw, (**b**) Municipal Selective Waste Collection Point (PSZOK) in Dąbrowa Białostocka.

**Figure 5 materials-13-05228-f005:**
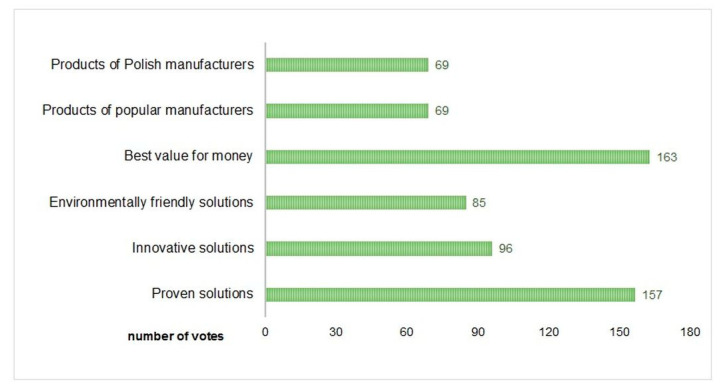
Decision-making factors in the purchase of construction products (multiple choice of respondents).

**Table 1 materials-13-05228-t001:** Content of the survey carried out among young- and middle-aged Poles.

No.	Question	Answer
1.	Gender	(a) Female; (b) Male; (c) Other
2.	Education	(a) Higher; (b) Secondary; (c) Primary; (d) Other
3.	Age	(a) Between 20–39; (b) Between 40–59; (c) Above 60
4.	Links with the construction sector	(a) Professional; (b) Employment in the construction sector; (c) People from the closest environment work in the construction sector; (d) No links; (e) Others
5.	How do you rate your knowledge and/or experience in the field of construction?	Scale from 1 to 5
6.	Have you ever participated in planning or carrying out construction works (e.g., supervising the renovation of an apartment, construction of a single-family house or elements of small architecture)?	(a) Yes; (b) Yes, I currently participate; (c) No, never
7.	How do you rate the need to protect the natural environment?	Scale from 1 to 5
8.	Please, assess your level of knowledge of the following terms: (1) sustainable development, (2) environmental protection, (3) sustainable development, (4) selective waste collection, (5) recycling, (4) Circular Economy	(a) I have very little information; (b) I have basic knowledge; (c) I know this topic quite well
9.	Which factors determine your choice when purchasing building materials?	(a) Proven solutions; (b) Innovative solutions; (c) Environmentally friendly solutions; (d) Best value for money; (e) Products of popular manufacturers; (f) Products of Polish manufacturers
10.	Please indicate 2 criteria that will determine your choice when purchasing gypsum plasterboards	(a) Price, (b) Dimension, (c) The origin of the plaster, (d) Manufacturer, (e) Availability of the product, (f) Other (please specify)
11.	Imagine you are supervising the construction of your own house on the property with a building to be demolish. When ordering the demolition service of this building, you will decide to:	(a) Basic service providing removal of CDW waste; (b) Premium service that ensures maximal re-use and recycling of construction and demolition waste (CDW) waste (ca. 20% more expensive than a)); (c) I prefer premium service to protect environment, but construction of the house is expensive so I will choose basic service; (d) It makes no difference to me, the most important thing is that the service should be provided immediately; (e) Other (please specify)

**Table 2 materials-13-05228-t002:** Characteristics of the Polish construction and assembly output in 2018 [10].

Aspect	Component	Share
Construction and assembly works	Specialized works	43.3%
	Construction of buildings	31.9%
	Construction of civil engineering structures	24.8%
Costs covered by medium and large enterprises (>9 employees)	Purchase of materials	46.5%
	Direct remuneration	14.1%
	Other direct costs	11.5%
	Other	27.9%
Exported construction works	Construction companies	89.7%
	Non-construction companies	10.3%
Construction works export countries (enterprises >9 employees)	Germany	48.6%
	Sweden	8.5%
	Belgium	7.5%
	Netherlands	4.5%
	Austria	4.2%
	Other countries	26.7%

**Table 3 materials-13-05228-t003:** Characteristics of Polish production related to construction [42].

Category of Products	Production in Bln EUR *	Number of Entities *	Representatives of the Category	Energy Consumed in kWh **	Share of Total Cost in % ***		
					Materials and Energy	External Services	Wages and Salaries
Products of wood, cork, straw and wicker	1.0	2152	sawmilling and planing	8.7	59.0	13.4	15.9
			wood, cork, straw products	11.1	65.7	12.7	11.7
Chemicals and chemical products	15.5	710	paints, varnishes and similar coatings, printing ink and mastics	1.6	67.8	11.7	11.4
Rubber and plastic products	22.4	2366	rubber products	5.9	64.3	13.1	13
			plastic products	6.8	66.0	11.7	12.7
Other non-metallic mineral products	14.1	1498	glass and glass products	11.8	55.2	16.6	15.6
			refractory products	4.7	66.6	12.8	12.3
			clay building materials	16.4	53.1	14.4	15.2
			cement, lime and plaster	31.4	44.2	30.4	9.2
			articles of concrete, cement and plaster	3.5	53.4	21.7	14.0
			abrasive products and non-metallic mineral products	13.0	59.1	18.7	12.0
Basic metals	13.5	437	pig iron, ferroalloys, steel and metallurgical products	18.5	77.6	11.8	5.1
			tubes, pipes, hollow profiles and related steel fittings	6.1	76.9	8.1	8.7
Metal products	28.7	5300	structural metal products	2.3	54.3	18.3	17.6

* Refers to the total number of entities operating within the same category of products in 2018; ** consumption of electricity per 23 EUR of sold production (refers to the selected representative in each category); *** data concerning economic entities keeping accounting ledgers (refers to the selected representative in each category).

**Table 4 materials-13-05228-t004:** Waste generated and landfilled (cumulative) so far in Poland [48].

Specification	Waste Generated in 2018						Waste Landfilled
	Grand Total	Recovered	Disposed ^1^		Transferred to Other Recipients ^2^	Temporarily Stored	
			Total	Of Which Landfilled ^1^			
	In Thousand Tonnes						
Total	115,339	58,429	54,636	49,046	1006	1267	1,760,072
Mining and quarrying	61,364	24,141	36,905	36,865	24	294	812,316
Manufacturing	26,117	19,485	5561	3051	444	627	266,359
Electricity, gas, steam and air conditioning supplies	18,341	8829	9171	8984	153	189	304,676
Water supply; sewerage waste management and remediation	5174	2147	2885	58	33	109	362,453
Construction	3774	3388	17	6	333	36	─
Other sections	569	439	97	83	19	13	14,268

^1^ Transferred to other recipients by the waste producer for recovery disposal (i.e., landfilling); ^2^ unknown direction of waste management.

**Table 5 materials-13-05228-t005:** Results of survey on knowledge of environmental protection and CE issues among Polish consumers.

Aspect	I have Very Little Information	I have Basic Knowledge	I Know this Topic Quite Well	Total Number of Votes
Sustainable construction in general	102	162	42	306
Environmental protection in general	8	76	222	306
Sustainable development	121	119	61	301
Selective collection of waste	21	165	118	304
Recycling	13	173	117	303
Circular Economy	136	117	45	298

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
