# Peer review of "Polish Transition towards Circular Economy: Materials Management and Implications for the Construction Sector"

_materials, 2020, doi:10.3390/ma13225228_

Round 1
Reviewer 1 Report
The issue raised in the paper "Polish transition towards circular economy: Materials management and implications for the Construction sector" it is of great interest at the present time that the construction sector is going through. Note that there has been a comprehensive study, very complete and consulting many sources and scientific research. Congratulations to the authors.
These comments and recommendations are made for the authors to consider in order for this work of such interest to be published in the journal:
_A review of the paper is proposed to the authors, to clarify what is the problem, the usefulness and the interest of this work for the readers. On the website of the Materials journal, it is stated with respect to “reviews” that: “These provide concise and precise updates on the latest progress made in a given area of research. Systematic reviews should follow the PRISMA guidelines”. The authors are recommended to follow the PRISMA guide, the checklist and the flow diagram to improve the focus of the paper.
_The paper could be completed by proposing the future lines of this research.
_Please, put in a clear way what is the interest of this paper.
_Structure of the paper: it is proposed to review the general structure of the paper.
_Line 9: Summary: please indicate what this paper contributes. What are the advances in this field?
_Tables and figures are not clear enough and are not well presented. See comments in the next section.
_Conclusions: Check that the conclusions that are provided are useful for the readers of the journal Materials.
_The bibliographic references are not complete and do not respect the format of the journal. It is appreciated that you review the format, thank you very much.
***
Other Comments:
_Keywords: please, check the format of the end point. How does this sentence end?
_Figure 1. Check text size. It is not readable, it is very small.
_Line 96-98: check the format of the scripts.
_Line 180 - 196. Check format. Check the separation between lines.
_Line 196: how are the references arranged in this journal ?. Please check these numbers: 8, 10 and 40. Are they ordered in order of appearance in the text? Check in the journal instructions.
_Line 261. How are these data for Poland valued relative to other countries?
_Figure 2 and 3. Check text size. It is not readable, it is very small.
_Figure 3: propose another type of graph. It is not appreciated well.
_The tables and figures are not sufficiently clear and are not well presented. Table 4. This table is not clear enough. It is difficult to understand which corresponds to each specification.
_Line 284 - 286: Check format. Check the separation between lines of the paragraph.
_Line 413: how is a figure cited in the text ?.
_Line 413: Review. There is no figure 5 in the paper.
_Line 457: check format.
_Line 465: references. Many of the references are incomplete, important information is missing. It is recommended to check the journal's instructions on how to cite a website, a journal, a book, the names of the authors, etc. For example: ref 5, 17, 19,46, 56, 58, etc.
_Line 465: references. Many of the references are not adapted to the format of the journal. It is recommended to check the journal's indications about the format of a website, a journal, a book, the names of the authors. For example: 4, 9, 21, 22, 23, 24, etc.
Author Response
Dear Reviewer, I sincerely thank you for your review and all comments which I have included in the revised version of the manuscript. Please find below my answer to your comments.
_The paper could be completed by proposing the future lines of this research.
I do agree with the Reviewer, the work should include information regarding the future lines of the research. The article in its actual shape, presents the current state of Polish Economy with special emphasis on construction sector and covers the beginning of the transformation process. Implementation of fundamental changes, such as the economic model, certainly deserves special attention, as it often implies reorientations in i.a. industrial as well as social trends, in the whole value chain. In consequence, detailed analysis of the effects of implemented changes, in regard to the particular branches of the economy and the social groups, as well as a global dimension of the transformation which directly impacts on the international politics of Poland should be conducted.
_Please, put in a clear way what is the interest of this paper. An improved interest of the paper has been indicated in the line 82.
_Structure of the paper: it is proposed to review the general structure of the paper. The article has been partially reorganised.
_Line 9: Summary: please indicate what this paper contributes. What are the advances in this field? The summary has been improved.
_Tables and figures are not clear enough and are not well presented. See comments in the next section. The figures and the tables were improved according to the comments included below.
_Conclusions: Check that the conclusions that are provided are useful for the readers of the journal Materials. The conclusion section has been improved and divided into 7 sections marked using bullet points.
_The bibliographic references are not complete and do not respect the format of the journal. It is appreciated that you review the format, thank you very much. The format of the references has been changed according to the journal’s requirements.
***
Other Comments:
_Keywords: please, check the format of the end point. How does this sentence end? The semicolon was removed.
_Figure 1. Check text size. It is not readable, it is very small. The text has been enlarged.
_Line 96-98: check the format of the scripts. The format of the scripts has been corrected.
_Line 180 - 196. Check format. Check the separation between lines. The separation between lines has been corrected.
_Line 196: how are the references arranged in this journal ?. Please check these numbers: 8, 10 and 40. Are they ordered in order of appearance in the text? Check in the journal instructions. The format of the references has been changed according to the journal’s requirements.
_Figure 2 and 3. Check text size. It is not readable, it is very small. The text of axis titles and tick labels has been enlarged.
_Figure 3: propose another type of graph. It is not appreciated well. The graph has been modified by introducing two types of tickles – circles and triangles which correspond to trade unit and production unit, respectively.
_The tables and figures are not sufficiently clear and are not well presented. Table 4. This table is not clear enough. It is difficult to understand which corresponds to each specification. The tables have been corrected by using alignment to left and additional edges of tables.
_Line 284 - 286: Check format. Check the separation between lines of the paragraph. The separation between lines has been corrected.
_Line 413: Review. There is no figure 5 in the paper. The figure 5 is placed on page 14 (line 433)
_Line 457: check format. 3.1. Subsection has been removed.
Reviewer 2 Report
I believe that the review article requires a more critical approach. Can more figures and tables be used to summarize the different findings/relationships among the construction sectors, materials, and circular economy (CE)? What are the parameters of the construction sector and materials are controlling the CE, etc.?
Y-axis tile including the unit of Figures 2 & 3 is missing.
To sum up, as an article for practicing engineers and policymakers, I rate the manuscript highly. Its scientific value is much lower, mainly due to the lack of criticism, a laconic summary in which the author did not indicate areas of knowledge that are unclear or insufficiently explored.
Author Response
Dear Reviewer, I sincerely thank you for your review and all comments which I have included in the revised version of the manuscript. Please find below my answer to your comments.
I believe that the review article requires a more critical approach. Can more figures and tables be used to summarize the different findings/relationships among the construction sectors, materials, and circular economy (CE)? What are the parameters of the construction sector and materials are controlling the CE, etc.? This work constitutes kind of brief compendium of knowledge regarding the current situation of Poland at the moment of the official beginning of economic transformation towards Circular economy model. At this moment nothing more than the parameters introduced by the EU in Monitoring framework for the circular economy COM(2018) 29 are used to monitor the progress of CE implementation.
Y-axis tile including the unit of Figures 2 & 3 is missing. The figures have been improved.
Reviewer 3 Report
The manuscript is well researched and well written.
Needs minor grammatical corrections.
Please move your computer cursor in the strike-through words with blue ink, you will see the corrected words/lines.

Author Response
Dear Reviewer, I sincerely thank you for your review and all comments which I have included in the revised version of the manuscript.
Reviewer 4 Report
The manuscript entitled "Polish transition towards circular economy: Materials management and implications for the construction sector” juxtaposed the Polish construction industry in the context of the national economy with the context of the evolving EU policies promoting green solutions. The resulting changes in Polish legislation, industry, and society are identified. The social aspects related to this transformation were analyzed based on a survey carried out among construction industry consumers.
This paper presents and summarizes good information about the Polish construction industry. However, few technical comments should be addressed before acceptance:
- This manuscript is more relevant to journals that focus on management and economy, not a journal of materials.
- The figures in this manuscript are unclear and should be improved to be more readable and clearer.
- The abbreviations should be defined at the first call in the manuscript and then they can be used.
- Line 284: The Figure should be “Figure 4”.
- Line 418: The Figure should be “Figure 5”.
- Line 457: What is the meaning of “1. Subsection” in the conclusion section.
- This reviewer recommends to re-phrase the conclusion section in a few points to be more specific.
Author Response
Dear Reviewer, I sincerely thank you for your review and all comments which I have included in the revised version of the manuscript. Please find below my answer to your comments.
The figures in this manuscript are unclear and should be improved to be more readable and clearer. This observation have shared other reviewers. The figures and tables have been significantly improved. I hope that current shape of the tables and figures meet expectations of the reviewers and future readers.
The abbreviations should be defined at the first call in the manuscript and then they can be used. A list of abbreviation has been created and placed after keywords.
Line 284: The Figure should be “Figure 4”. It has been corrected.
Line 418: The Figure should be “Figure 5”. It has been corrected.
Line 457: What is the meaning of “1. Subsection” in the conclusion section. “1. Subsection” in at the end of the conclusion section was a mistake.
This reviewer recommends to re-phrase the conclusion section in a few points to be more specific. The conclusion section have been divided into 7 sections marked using bullet points.
Round 2
Reviewer 1 Report
The work of the authors in improving the paper “"Polish transition towards circular economy: Materials management and implications for the construction sector” is appreciated. The authors have considered the reviewer’s recommendations and the improvement of the paper has been important.
The authors have improved the abstract, new contributions, conclusions, references, figures, tables, etc.
Please a revision of the format of all the tables, figures and text in this new version is necessary.
Author Response
Dear Reviewer,
the tables, figures and text have been revise.
I would like to thank you again for all comments which definitely helped to improve the article.
Kind regards,
Justyna Tomaszewska
Reviewer 2 Report
I have no more comments
Author Response
Dear Reviewer,
I would like to thank you again for all comments which definitely helped to improve the article.
Kind regards,
Justyna Tomaszewska